# Optimization of FFF Process Parameters by Naked Mole-Rat Algorithms with Enhanced Exploration and Exploitation Capabilities

**DOI:** 10.3390/polym13111702

**Published:** 2021-05-23

**Authors:** Jasgurpreet Singh Chohan, Nitin Mittal, Raman Kumar, Sandeep Singh, Shubham Sharma, Shashi Prakash Dwivedi, Ambuj Saxena, Somnath Chattopadhyaya, Rushdan A. Ilyas, Chi Hieu Le, Szymon Wojciechowski

**Affiliations:** 1Department of Mechanical Engineering, Chandigarh University, Mohali 140413, India; jasgurpreet.me@cumail.in (J.S.C.); ramankakkar@gmail.com (R.K.); 2Department of Electronics and Communication Engineering, Chandigarh University, Mohali 140413, India; mittal.nitin84@gmail.com; 3Department of Civil Engineering, Chandigarh University, Mohali 140413, India; drsandeep1786@gmail.com; 4Department of Mechanical Engineering, IK Gujral Punjab Technical University, Main Campus, Kapurthala 144603, India; 5G.L. Bajaj Institute of Technology & Management, Greater Noida 201310, Gautam Buddha Nagar, India; spdglb@gmail.com (S.P.D.); ambuj.saxena1@gmail.com (A.S.); 6Department of Mechanical Engineering, Indian Institute of Technology (ISM), Dhanbad 826004, Jharkhand, India; somnathchattopadhyaya@iitism.ac.in; 7Faculty of Engineering, School of Chemical and Energy Engineering, Universiti Teknologi Malaysia, Johor Bahru 81310, Malaysia; ahmadilyas@utm.my; 8Centre for Advanced Composite Materials, Universiti Teknologi Malaysia, Johor Bahru 81310, Johor, Malaysia; 9Faculty of Engineering and Science, University of Greenwich, Kent ME4 4TB, UK; c.h.le@gre.ac.uk; 10Faculty of Mechanical Engineering and Management, Poznan University of Technology, 60-965 Poznan, Poland

**Keywords:** fused-filament fabrication, mechanical strength, naked mole-rat algorithm, optimization, process parameters

## Abstract

Fused filament fabrication (FFF) has numerous process parameters that influence the mechanical strength of parts. Hence, many optimization studies are performed using conventional tools and algorithms. Although studies have also been performed using advanced algorithms, limited research has been reported in which variants of the naked mole-rat algorithm (NMRA) are implemented for solving the optimization issues of manufacturing processes. This study was performed to scrutinize optimum parameters and their levels to attain maximum impact strength, flexural strength and tensile strength based on five different FFF process parameters. The algorithm yielded better results than other studies and successfully achieved a maximum response, which may be helpful to enhance the mechanical strength of FFF parts. The study opens a plethora of research prospects for implementing NMRA in manufacturing. Moreover, the findings may help identify critical parametric levels for the fabrication of customized products at the commercial level and help to attain the objectives of Industry 4.0.

## 1. Introduction

In the last few decades, the manufacturing sector has witnessed a paradigm shift from conventional to advanced manufacturing techniques in both developed and developing nations. Another breakthrough was achieved by researchers after developing additive manufacturing (AM) technologies, which is gradually taking the present industry into the era of Industry 4.0 [1]. The ever-increasing demand for good quality and customized products at lower cost has brought additive manufacturing technologies into the limelight. AM primarily solves the problem of long delay periods between customer demands and product development. During new product development, conventionally, it must pass through many stages, which are eliminated in AM, even for complex parts [2]. This technology used digital fabrication methods where the computer-generated design of the required product is created through a layer-by-layer manufacturing technique. There is no requirement for jigs, fixtures, dies and other tools for processing. Researchers have even successfully developed multiple valve attachments for ventilators during emergency conditions, such as pandemic occurred in 2020 [3]. Thus, the rapid delivery of high-quality customized products while maintaining minimum cost is the reason for the supremacy of these techniques.

Fused-filament fabrication (FFF) is the most popular AM technology, which works on the principle of layer-by-layer manufacturing as designated by ASTM [4]. This technology can transfer conceptual digital drawing into an actual product within a few hours using a wide range of polymers and polymer composite filaments as raw material. The fiber and particle-reinforced composites find vast applications in the areas of medicine, aerospace, automobile and electronics [5,6]. Moreover, carbon-reinforced composites fabricated using FFF are more cost-efficient than powdered alloys used in metal printers [7]. Due to its material flexibility, easy transportation, minimal environmental degradation, lower installation and material cost [8], researchers have used it extensively while developing smart materials, metamaterials, implants, and other biomaterials for medical applications. Medical 3D printing is gaining popularity as customized anatomical models are prepared with higher accuracy and minimum lead time [9]. The inclusion of materials with sensing capabilities in 3D printing has realized the dream of producing smart materials [10]. FFF uses two-dimensional motion (X- and Y-direction) of heated extrusion nozzle, moved by stepper motors and controlled by a microprocessor. The feedstock filament is passed by rotating rollers into the extrusion nozzle, which heats it slightly lower than the melting point, as shown in Figure 1. This semi-molten polymer bead is deposited on the build platform; after one layer is deposited, the build platform moves downward (Z-direction), and the next layer of materials is deposited. The motion of the nozzle and build platform are synchronized by the microprocessor according to the product design [11]. The part is immediately cooled and cleaned after completion, after which it can be used for the desired application.

Despite advantages, FFF has certain process limitations like poor surface finish and mechanical strength than injection molding [12]. The surface finish can be enhanced by postprocessing methods, including mechanical finishing, chemical finishing and vapor smoothing. However, there is no method available, which can improve the mechanical strength of FFF parts. This aspect limits the usability of these components for certain applications where flexural strength, compressive strength and tensile strength are mandatory [13]. On the contrary to postprocessing, the FFF part shows significant variation in mechanical stability while manufactured at different conditions. There are numerous studies, which have found that process parameters like layer thickness, raster angle, build temperature, orientation angle, infill density, infill pattern and layer gap have a significant impact on the mechanical stability of FFF parts [14,15]. As a general notion, the filament layers deposited in the direction of tensile force and compressive force results in higher mechanical strength [11]. Honeycomb structure with 100% infill density has yielded maximum tensile strength, as reported by Fernandez-Vicente et al. [16]. However, complex issues arise when estimating strength is done for intricate designs manufactured using different build strategies. In addition to conventional optimization techniques, researchers have utilized advanced algorithms to identify optimum parameter settings to attain higher mechanical stability. Section 2 presents different optimization algorithms used for mechanical strength enhancement of FFF parts. Afterward, in Section 3 and Section 4, the novel naked mole-rat algorithm (NMRA) and its variants are discussed. Section 5 elaborates the case study performed to test the performance of advanced algorithms followed by simulation results and discussion in Section 6.

## 2. Literature Review

Most of the studies, which reported using advanced algorithms for the FFF process were performed to enhance surface finish and mechanical strength of parts by optimizing various process parameters. Initially, conventional techniques were used for optimizing tensile strength based on orientation angle. At the same time, results were compared with finite element analysis (FEA) data. The simulated results were in strong correlation with experimental data, thus validating the FEA model. The maximum tensile strength has occurred at a 45° orientation angle [17]. Panda et al. [18] studied the impact of several process parameters of FFF-like layer thickness, raster width, raster angle, orientation, and air gap on tensile strength, flexural strength and impact strength of standard test samples. Experiments were carried out, and bacterial foraging technique was implemented after ANOVA tests. This advanced optimization technique was robust enough to predict the response for parameters outside the range of mathematical models formulated by experimental data.

Rayegani and Onwubolu [19] proposed a relation between input parameters of FFF with tensile strength using the hybrid group method of data handling. The raster angle, air gap, orientation angle, and raster width varied during experimentation. Results were compared with modeled data, which showed a strong correlation between experimental and predicted tensile strength. Goudswaard et al. [20] tested and compared the efficacy of evolutionary algorithm, particle swarm optimization and Simulated Annealing during optimizing tensile strength of FFF parts. It was clear that particle swarm optimization outclassed both evolutionary and simulated annealing algorithms in quality and consistency. Liu et al. [13] solved the problems of the anisotropic behavior of FFF materials using hybrid deposition path planning and topology optimization technique. The anisotropy is induced by tool paths generated during the slicing of part, which results in the variation of mechanical properties in different loading directions. The hybrid tool path was developed using the techniques above, which performed better than regular infill strategies, such as crisscross and contour offset. The simulated results were compared with actual parts fabricated using a new algorithm, which validated its efficacy.

Genetic programming technique [1] was used for optimizing fatigue behavior of FFF parts considering six input parameters with three levels of each. Contour number was found to be a significant parameter than the other five. At the same time, genetic programming outperformed response surface methodology during the prediction of fatigue.

The artificial neural networks technique was successfully implemented to enhance the creep behaviors of FFF parts after investigating the impact of five input parameters [21]. The best creep compliance occurred at 0.127 mm layer thickness, zero raster angle and air gap, 17.188˚ deposition angle, 0.4572 mm width with 10 contours. ANN emerged as the most popular optimization tool for manufacturing processes. It can be applied to nonlinear and multitudinous cases, which was validated by this study. Raju et al. [22] utilized hybrid particle swarm and bacterial foraging techniques for four FFF process parameter optimization while studying their impact on hardness, mechanical strength and surface finish of test samples. Compared to conventional optimization tools, the hybrid tool yielded a 7.44% higher response in all the output parameters. Moreover, this tool also performed better than the other algorithms like conventional particle swarm and bacterial foraging. Rao and Rai [23] solved the single and multiobjective optimization issues of FFF using simple and non-dominated TLBO techniques, respectively. Five case studies were discussed with different objective functions and parameter bounds. The response parameters, such as compressive strength, sliding wear, flexural strength, impact strength and tensile strength, were investigated using advanced optimization tools. All the mathematical models were created by previous researchers during experimentations, which were employed as input to study and compare the performance of the optimization tools, which proves highly efficient than previous results.

Malviya and Desai [24] coupled two techniques, i.e., artificial neural networks and Bayesian algorithm, to optimize orientation angle. A significant impact on tensile strength parameters was experienced. The proposed methodology has successfully achieved desired goals despite the anisotropic behavior of FFF parts. Yadav et al. [25] discussed the impact of infill density, material density and nozzle temperature on the tensile strength of FFF parts manufactured by different materials. They used a hybrid genetic algorithm-artificial neural network technique for the optimization. The experimental data showed strong similarity with predicted results with an error of less than 3%, thus validating the efficiency of the hybrid algorithm. Natarajan et al. [26] used a Non-dominated sorting modified TLBO for conventional machining processes. The impact of four input parameters was studied during the machining of polytetrafluoroethylene with a cemented carbide tool. This algorithm performed better than the other six algorithms by yielding more uniform results and non-dominated solutions. While comparing predicted and experimental results, only 3.7% error was observed while validating the efficacy of this tool. Some researchers also used different algorithms to solve a single objective function to compare and identify the most efficient algorithm. Saad et al. [27] tested four different algorithms for minimizing the surface roughness of FFF parts. The symbiotic organism search algorithms were most effective as minimum surface roughness was predicted at optimum parameter settings, further validated by experimental results.

Another research reported using the firefly algorithm to optimize the wear rate of copper-plated FFF parts. The mathematical model was designed based on the raster angle, air gap, voltage and time to predict the wear rate [28]. A Multicriteria genetic algorithm was utilized by Pandey et al. [2] to predict optimum orientation angle against two objective functions, i.e., build time and surface roughness. The algorithm was robust enough to predict the surface roughness and build time for parts of any geometry.

The research, development and testing of the efficacy of new algorithms is a never-ending process as advanced versions of algorithms are rapidly developed by researchers. This study focuses on optimizing process parameters of FFF to attain maximum mechanical strength using NMRA [29] and its variants, which are discussed in the next section.

## 3. Naked Mole-Rat Algorithm Variants

Researchers are motivated by the behavioral characteristics of naked mole rats (NMRs) to design an optimization technique termed NMRA [29]. The algorithm reproduces the natural mating patterns of NMRs for designing algorithms. NMRs are classified into two types, i.e., breeders and worker NMRs. The breeders are the most efficient NMRs in the group, and these are only intended for mating with the queen. Workers are doing the additional tasks, and the best performing workers will move on to the breeding group. The best executor of the breeder’s pool will be the queen’s mating partner.

There are three steps to the algorithm. The NMR group is initially created, and the NMR group is then divided into employees and breeders. The group of breeders is selected based on the probability of breeding. The NMRA steps are as follows:

### 3.1. Initialization

Initially, n NMRs are generated at random. The D-dimensional vector space is represented by each NMR in the range of [1,2,… …,n]. Each NMR shall be initialized as
(1)$$NMR_{i,j}=NMR_{min,j}+U\left(0,1\right)\times \left(NMR_{min,j}-NMR_{max,j}\right)$$
where i∈[1, 2,… …,n], j∈[1, 2,… …,D], and U(0,1) is a uniform random number. After initialization, the objective function and fitness values are determined, and breeders (B) and workers (W) based on their fitness function and the best solution d is selected.

### 3.2. Worker Phase

To switch to the breeder group and to get a chance to mate with the queen, each worker tries to improve his fitness. The new fitness value of NMR is calculated, and the updated solution is selected and recorded if the fitness is better than the previous one. The previous solution will otherwise be chosen.

The equation used to generate the updated solution is as follows: (2)$$w_{i}^{t+1}=w_{i}^{t}+\lambda \left(w_{j}^{t}-w_{k}^{t}\right)$$
where ith worker is represented by $w_{i}^{t}$ in the tth iteration. The random solution (jth and kth worker) chosen from the worker’s group is $w_{j}^{t}$ and $w_{k}^{t}$. $w_{i}^{t+1}$ represents a new fitness solution. A uniform distribution ranges from [0, 1] provides the value of λ.

### 3.3. Breeder Phase

The breeder needs to update the NMRs to be chosen as their partner or to remain a breeder. The breeder’s NMRs are updated with the best d overall reproductive probability ($b_{p}$). This $b_{p}$ is a random [0, 1] number. If breeders do not update fitness, they may be removed from the worker’s pool:(3)$$b_{i}^{t+1}=\left(1-\lambda \right)b_{i}^{t}+\lambda \left(d-b_{i}^{t}\right)$$
where in tth iteration $b_{i}^{t}$ represents ith breeder, λ controls the mating frequency and assists in the next iteration in identifying a new breeder $b_{i}^{t+1}$. At first, $b_{p}$ is set to 0.5 for breeders.

The entire search process is repeated iteratively until the termination criteria is satisfied. Consequently, the best breeder chosen from the whole population is the possible solution to the problem under study.
**Algorithm 1** Pseudocode of NMRA
***Input:*****Define objective function*****f*****(NMR), NMR = (NMR1, NMR2, . . ., NMR****_*D*_)*****Output:*****Identify current best solution**d**;*****Initialization:*****Initialize NMRs: n, breeders*****B*****:*****n*****/****5****, workers*****W*****:*****B*****−*****n***             **Describe breeding probability:**
***bp***
       ***while iterations <***
$it_{max}$
***for i = 1: w*** $w_{i}^{t+1}=w_{i}^{t}+\lambda \left(w_{j}^{t}-w_{k}^{t}\right)$
***evaluate***$w_{i}^{t+1}$           ***end for***
***for i = 1: B******if rand* >*****bp*** $b_{i}^{t+1}=\left(1-\lambda \right)b_{i}^{t}+\lambda \left(d-b_{i}^{t}\right)$
***end if******evaluate***$b_{i}^{t+1}$***end for******combine the updated breeder and worker population******estimate the NMRs******update the overall best d******update iteration count******end while******update final best***d***end***

Because of its linear nature, NMRA has recently gained interest among researchers. The revised NMRA improves performance by improving both its basic exploitation and exploration capabilities. The elite opposition-based learning (EOBL) strategy [30] improves the exploration of basic NMRA. Exploitation is improved by local neighborhood search (LNS) with information on the best solution to date in the small neighborhood of the solution [25]. The following are the main changes:

#### 3.3.1. NMRA Version 1.0

NMRV 1.0 has improved NMRA’s exploration capabilities through implementing the EOBL Strategy [30]. The opposition-based NMRA, also called NMRV 1.0, Algorithm 2 detailed the pseudocode of NMRV 1.0.
**Algorithm 2** Pseudocode of NMRV 1.0
***Input:*****Define objective function*****f*****(NMR), NMR = (NMR1, NMR2, . . ., NMR*****_D_*****)*****Output:*****Identify current best solution**d**;*****Initialization:*****Initialize NMRs: n, breeders*****B*****:*****n*****/****5****, workers*****W*****:*****B*****−*****n***              **Describe breeding probability:**
***bp***
          **Update the current population with EOBL;**
***while iterations <***$it_{max}$***for i = 1: w*** $w_{i}^{t+1}=w_{i}^{t}+\lambda \left(w_{j}^{t}-w_{k}^{t}\right)$
***evaluate***$w_{i}^{t+1}$              ***end for***
***for i = 1: B******if rand* >*****bp*** $b_{i}^{t+1}=\left(1-\lambda \right)b_{i}^{t}+\lambda \left(d-b_{i}^{t}\right)$
***end if******evaluate***$b_{i}^{t+1}$***end for*****c*ombine the updated breeder and worker population******estimate the NMRs******update the overall best d******update iteration count******end while******update final best***d***end***

#### 3.3.2. NMRA Version 2.0

The local neighborhood search (LNS) model has been employed [30] to enhance the exploitation capacity of NMRA. In the basic NMRA, workers change their position during the employee phase according to local information and their past experience. The new best solution is restructured with LNS in the upgraded NMRA version 2.0, and the worker phase is updated as a solution:(4)$$L_{i}^{t}=w_{i}^{t}+u*\left(w_{n\_opt}-w_{i}^{t}\right)+v*\left(w_{p}^{t}-w_{q}^{t}\right)$$
where $w_{n\_opt}$ is the best solution in the neighborhood of $w_{i}^{t}$ and u, v ∈ rand() are the scaling factors, and p, q ∈ [*i* − *r*, *i* + *r*] (*p*
≠
*q*
≠
*i*) neighborhood. The updated solution using LNS in the worker phase is given by:(5)$$w_{i}^{t+1}=L_{i}^{t}+\lambda \left(w_{j}^{t}-w_{k}^{t}\right)$$
where λ ∈ rand( ) is a scaling factor, $L_{i}^{t}$ is the LNS updated best solution. The pseudocode of NMRV 2.0, i.e., LNS-based NMRA, is shown in Algorithm 3.
**Algorithm** 3 Pseudocode of NMRV 2.0
***Input:*****Define objective function*****f*****(NMR), NMR = (NMR1, NMR2, . . ., NMR*****_D_*****)*****Output:*****Identify current best solution**d**;*****Initialization:*****Initialize NMRs: n, breeders*****B*****:*****n*****/*****5*****, workers*****W*****:*****B*****−*****n***              **Describe breeding probability:**
***bp***
          ***while iterations <***
$it_{max}$
***for i = 1: w***$L_{i}^{t}=w_{i}^{t}+u*\left(w_{n\_opt}-w_{i}^{t}\right)+v*\left(w_{p}^{t}-w_{q}^{t}\right)$$w_{i}^{t+1}=L_{i}^{t}+\lambda \left(w_{j}^{t}-w_{k}^{t}\right)$***evaluate***$w_{i}^{t+1}$              ***end for***
***for i = 1: B******if rand* >*****bp***$b_{i}^{t+1}=\left(1-\lambda \right)b_{i}^{t}+\lambda \left(d-b_{i}^{t}\right)$***end if******evaluate***$b_{i}^{t+1}$***end for*****c*ombine the updated breeder and worker population*****estimate the NMRs****u*pdate the overall best d******update iteration count******end while******update final best***d***end***

#### 3.3.3. NMRA Version 3.0

NMRA version 3.0 has two modifications in basic NMRA to balance the exploitation and exploration capacity [30]. First, the exploration tendency is improved by the EOBL strategy. Second, the exploitation capability is improved by LNS. The pseudocode of NMRV 3.0 is given in Algorithm 4.
**Algorithm 4** Pseudocode of NMRV 3.0
***Input:*****Define objective function*****f******(NMR), NMR = (NMR1, NMR2, . . ., NMR*****_D_****)*****Output:*****Identify current best solution**d**;*****Initialization:*****Initialize NMRs: n, breeders*****B*****:*****n*****/****5****, *workers******W*****:*****B******−******n***             **Describe breeding probability:**
***bp***
        ***Update the current population with EOBL;***
****while iterations* <***$it_{max}$***for i = 1: w*** $L_{i}^{t}=w_{i}^{t}+u*\left(w_{n\_opt}-w_{i}^{t}\right)+v*\left(w_{p}^{t}-w_{q}^{t}\right)$
                $w_{i}^{t+1}=L_{i}^{t}+\lambda \left(w_{j}^{t}-w_{k}^{t}\right)$
***evaluate]***$w_{i}^{t+1}$             ***end for***
***for i = 1: B******if rand* >*****bp*** $b_{i}^{t+1}=\left(1-\lambda \right)b_{i}^{t}+\lambda \left(d-b_{i}^{t}\right)$
***end if******evaluate***$b_{i}^{t+1}$***end for******combine the updated breeder and worker population******estimate the NMRs******update the overall best d******update iteration count******end while******update final best***d***end***

## 4. Case Study

The simulation results were compared with the optimization study carried out by Panda et al. [18], where the influence of five FFF parameters was optimized to improve the mechanical properties. The process parameters utilized for this study are:$$x_{1}\;Layer\;thickness\;\left(in\;mm\right)\;x_{2}\;Building\;orientation\;\left(in\;degree\right)\;x_{3}\;Raster\;angle\;\left(in\;degree\right)\;x_{4}\;Raster\;width\;\left(in\;mm\right)\;x_{5}\;Air\;gap\;\left(in\;mm\right)$$

The primary motivation for selecting the above-mentioned input parameters is the literature review, which has found a good correlation between abovesaid parameters with mechanical properties of FFF components [2,19]. The output variables used for this case study are impact strength (IS) in $MJ/m^{2}$, flexural strength (FS) in MPa and Tensile Strength (TS) in MPa. The main reason for selecting these output parameters is that these components undergo different types of loading conditions for medical, aerospace and automobile applications. Therefore, it is mandatory to evaluate all these mechanical properties utilizing an advanced optimization algorithm. These response parameters are considered as an objective function for the present case study.

The objective function is shown in Equations (6)–(8):(6)$$TS=13.5625+0.7156x_{1}-1.3123x_{2}+0.9760x_{3}+0.5183x_{5}+1.1671x_{1}^{2}-1.3014x_{2}^{2}-0.4363x_{1}x_{3}\;+0.4364x_{1}x_{4}-0.4364x_{1}x_{5}+0.4364x_{2}x_{3}+0.4898x_{2}x_{5}-0.5389x_{3}x_{4}+0.5389x_{3}x_{5}-0.5389x_{4}x_{5}$$
(7)$$FS=29.9178+0.8719x_{1}-4.8741x_{2}+2.4251x_{3}-0.9096x_{4}+1.6626x_{5}-1.7199x_{1}x_{3}+1.7412x_{1}x_{4}\;-1.1275x_{1}x_{5}+1.0621x_{2}x_{5}+1.0621x_{3}x_{5}+1.0408x_{4}x_{5}$$
(8)$$IS=0.401992+0.034198x_{1}+0.008356x_{2}+0.013673x_{3}+0.021383x_{1}^{2}+0.008077x_{2}x_{4}$$

It is required to maximize each type of strength; thus, maximization of objective functions is performed.

The lower and upper bounds of process parameters are identified based on constraints as most commercially available FFF machines can manufacture components within the following parameter bounds. The parameter bounds for objective function are expressed in Table 1 as follows:

## 5. Simulation Results and Discussion

In the FFF process, to resolve this optimization problem, the NMRA variants are utilized. Since heuristic algorithms are stochastic optimization methods, these must be executed at least 10 times to generate meaningful statistics. For this purpose, the simulations are performed 30 times. Population sizes vary from 20 to 40, and the number of iterations varies from 100 to 500. The NMRA variants have been selected for efficiency check by the FFA [31], FPA [32], NBA [33], SCA [34] and SSA [35] algorithms. Table 2 details the parameter settings used to compare the results.

Here, NP is the population size, dim is the dimension of the problem, $it_{max}$ is the number of iterations

The optimum results attained by simulated algorithms with population size 40 and maximum amount of iterations, i.e., 500, are shown in Table 3, Table 4 and Table 5 for *TS, FS* and *IS,* respectively. It is clear from the results that the NMRV 2.0 and 3.0 algorithm’s fitness values are slightly better than others for *TS* and *FS* fitness functions, respectively. Outcomes of all competitive algorithm’s fitness values are nearly similar to competitive algorithms for *IS* fitness function. The convergence rate for *TS, FS* and *IS* are drawn in Figure 2, Figure 3 and Figure 4, respectively, which is also better than others. The performance of simulated algorithms for FFF for *TS, FS* and *IS* are specified in Table 3, Table 4 and Table 5, respectively. This demonstrates that NMRV 3.0 algorithm’s standard deviation is better than others that demonstrates improved exploitation and exploration capabilities of NMRV 3.0 for process parameters optimization of FFF as confirmed by box-plots shown in Figure 5, Figure 6 and Figure 7.

The performance evaluation of simulated algorithms with population size 40 and maximum amount of iterations, i.e., 100, are specified in Table 6, Table 7 and Table 8 for *TS, FS* and *IS,* respectively. It is clear from the statistics that NMRV 3.0 algorithm’s fitness values are slightly better than others for *TS, FS* and *IS* fitness functions.

To further check the effectiveness of proposed variants, the simulations were carried out for different numbers of iterations and population sizes [36]. The performance evaluation of simulated algorithms with population size 20 and maximum amount of iterations, i.e., 100, are given in Table 9, Table 10 and Table 11. It is clear from the results that NMRV 3.0 algorithm’s fitness values are slightly better than others for *TS, FS* and *IS* fitness functions.

## 6. Confirmatory Experiments

To validate the findings of NMRA, the confirmatory experiments were planned to fabricate sample parts at optimized parameters followed by mechanical testing. Five samples were manufactured using the FFF process at predefined conditions as suggested by NMRA. The samples are prepared for confirmation of *TS* and *FS* of parts fabricated at optimum parameter settings. The samples for *TS* and *FS* are prepared as per ASTM D638 and ASTM D790 standards. The dimensions of *TS* and *FS* samples are shown in Figure 8.

Acrylonitrile butadiene styrene (ABS) has been extensively used for automobile and electronic components and hence it is most popular raw material used in FFF printers. The filament material selected for the fabrication of test samples in the present study is ABS P400 with a 1.75 mm diameter supplied by Robokits, India. Table 12 displays the FFF process parameters selected during the fabrication of test samples.

The FFF machine used for the fabrication of test parts was Model I3 supplied by Prusa Research, Prague, Czech Republic and Universal testing machine (UTM) was supplied by Shanta Engineering Pvt. Ltd., Pune, India, as shown in Figure 9. The UTM was operated at 50 mm/min strain rate with gradual bending load subjected at three points during flexural tests (see Figure 9d). During tensile testing, the samples were held in two jaws with 52 mm grip separation. The tensile force was exerted gradually at 50 mm/min until the samples were fractured. Afterward, the fracture points of both tensile and flexural test samples were studied using scanning electron microscope (SEM) images which were generated by Model IT500HR supplied by Jeol Ltd. Tokyo, Japan.

The mechanical strength of parts was measured and compared with simulated results, as shown in Figure 10. The experimental results were compared with predictions made by the NMRA algorithm at suggested parametric settings. Notably, experimental results were in significant agreement with the modeled data when experiments were performed at optimum parameter settings. The output of NMRA yielded significant and promising results, which may be beneficial for deciding 3D printing and part manufacturing conditions at the commercial level. During mass production and part production, the optimized parametric settings would be utilized to attain desired mechanical properties in products.

In addition, the SEM images retrieved after mechanical testing reveal the conditions at the location of breakage. Figure 11a shows the SEM micrographs of tensile test parts, indicating higher flexibility in the layers at the point of failure. A higher elongation was experienced for these parts as an optimized combination of parameters was used. In Figure 9b, the flexural test sample was photographed. It can be confirmed that failure occurs after extreme elongation due to flexibility induced in parts. The mechanical strength and flexibility attained at this location reveal the strategic selection of parametric levels.

## 7. Conclusions and Future Scope

The mechanical strength of FFF parts is significantly lower than conventionally prepare thermoplastics components, which hinders their applicability for certain applications. Thus, we must identify optimum process parameters to enhance *TS, FS* and *IS* of FFF parts. The performance of advanced naked mole-rat algorithm variants was tested to solve FFF issues of poor mechanical strength. The results indicated a significant enhancement of tensile, flexural and impact strength than previous studies using NMRV 3.0. The study could be further extended to identify optimum parameter settings to achieve maximum dimensional accuracy and surface finish of FFF components. Moreover, the efficacy of the enhanced versions of NMRA algorithms could be further tested for optimizing parameters of the other additive manufacturing techniques, such as stereolithography, electron beam melting and selective laser melting.

## Figures and Tables

**Figure 1 polymers-13-01702-f001:**
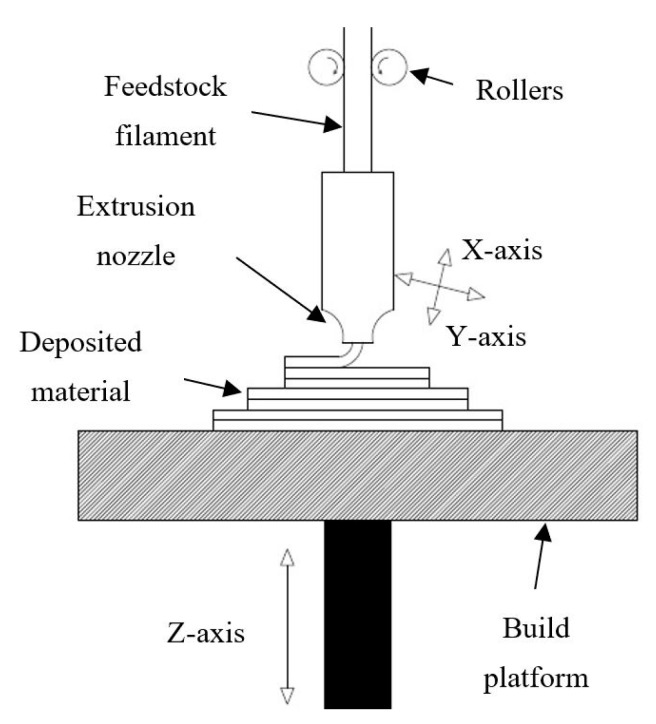
Schematic of the Fused Filament Fabrication (FFF) process.

**Figure 2 polymers-13-01702-f002:**
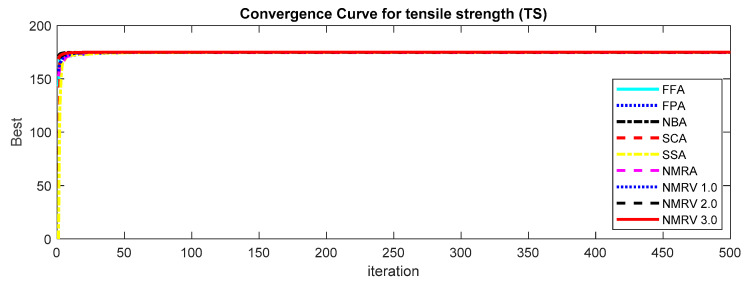
Convergence graph of simulated algorithms with population size 40 over 30 runs for 500 iterations for Tensile Strength (*TS)* estimation.

**Figure 3 polymers-13-01702-f003:**
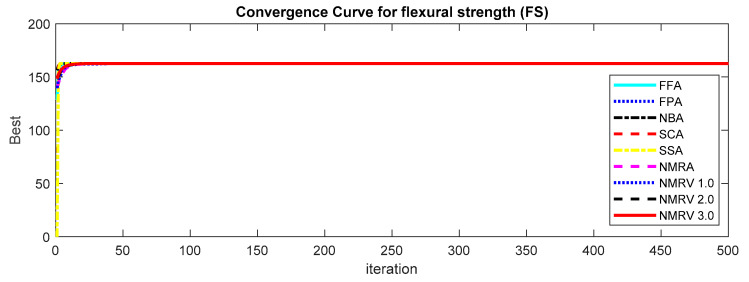
Convergence graph of simulated algorithms with population size 40 over 30 runs for 500 iterations for Flexural Strength (*FS)* estimation.

**Figure 4 polymers-13-01702-f004:**
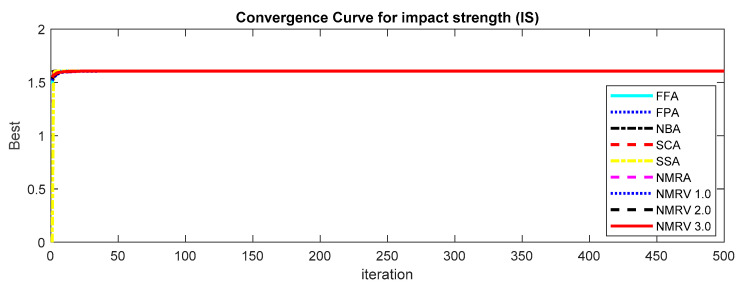
Convergence graph of simulated algorithms with population size 40 over 30 runs for 500 iterations for Impact Strength (*IS)* estimation.

**Figure 5 polymers-13-01702-f005:**
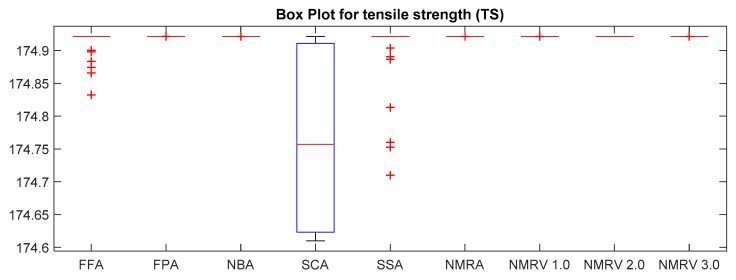
Box plot of simulated algorithms with population size 40 over 30 runs for 500 iterations for *TS* estimation.

**Figure 6 polymers-13-01702-f006:**
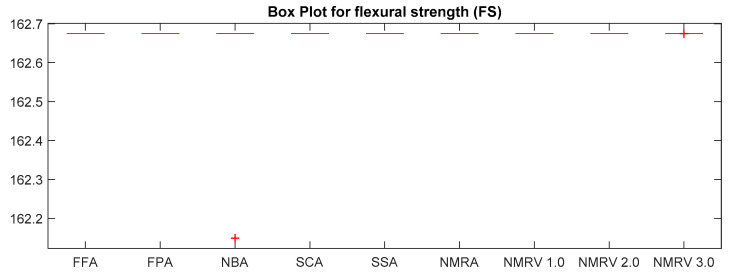
Box plot of simulated algorithms with population size 40 over 30 independent runs for 500 iterations for *FS* estimation.

**Figure 7 polymers-13-01702-f007:**
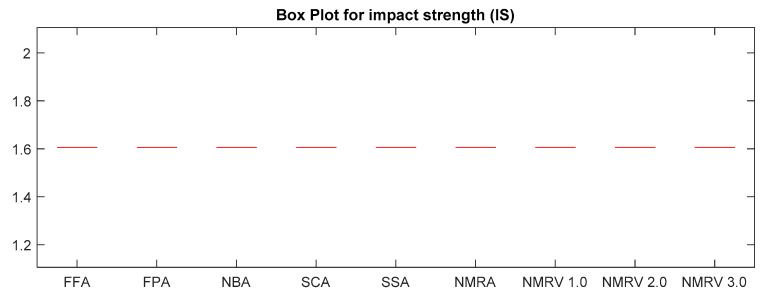
Box plot of simulated algorithms with population size 40 over 30 independent runs for 500 iterations for *IS* estimation.

**Figure 8 polymers-13-01702-f008:**
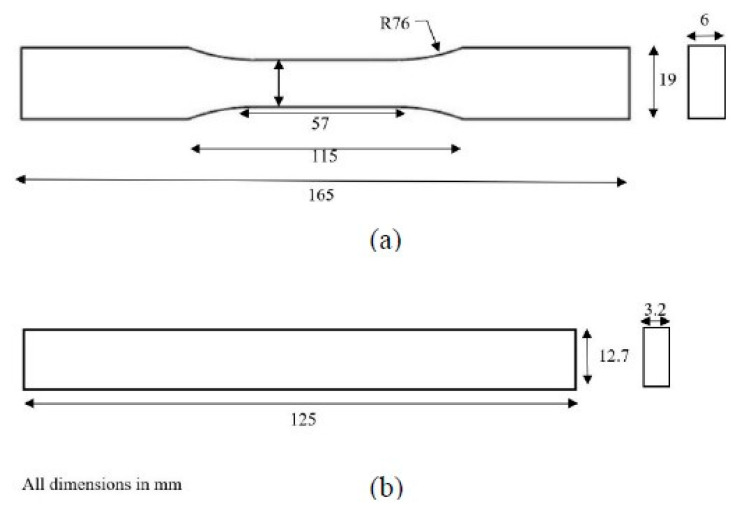
Dimensions of test parts for (**a**) *TS* (**b**) *FS*.

**Figure 9 polymers-13-01702-f009:**
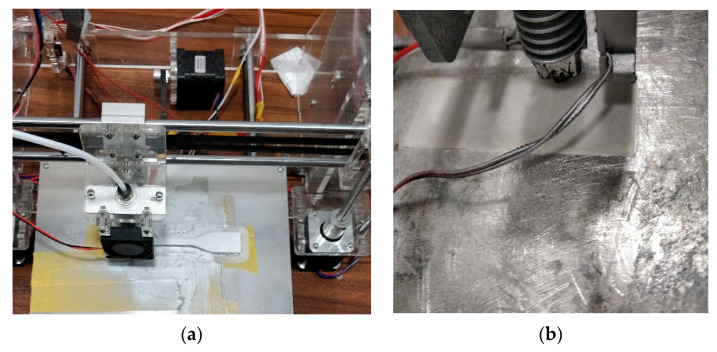
(**a**) FFF printer (**b**) *FS* sample during the fabrication (**c**) Universal Testing Machine (UTM) used for testing (**d**) *FS* sample during testing.

**Figure 10 polymers-13-01702-f010:**
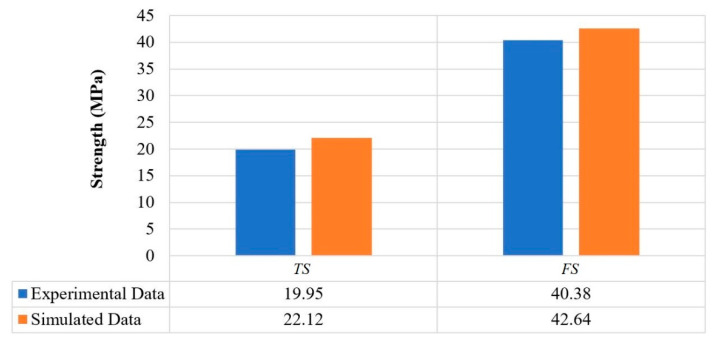
Comparison of experiments and simulated results of mechanical strength.

**Figure 11 polymers-13-01702-f011:**
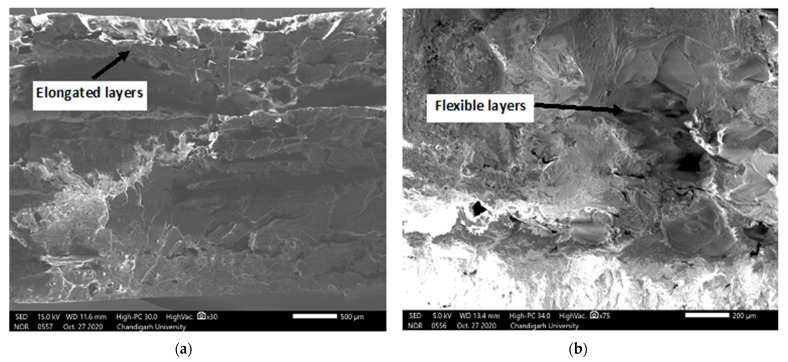
SEM images of point of failure of parts during (**a**) tensile test (**b**) flexural test.

**Table 1 polymers-13-01702-t001:** Parameter bounds for the objective function.

Parameter	Lower Bound	Upper Bound
$x_{1}\;Layer\;thickness\;\left(in\;mm\right)$	0.127	0.254
$x_{2}\;Building\;orientation\;\left(in\;degree\right)$	0	30
$x_{3}\;Raster\;angle\;\left(in\;degree\right)$	0	60
$x_{4}\;Raster\;width\;\left(in\;mm\right)$	0.4064	0.5064
$x_{5}\;Air\;gap\;\left(in\;mm\right)$	0	0.008

**Table 2 polymers-13-01702-t002:** Parameter values.

Algorithm	Parameters
FFA	$NP=20-40,\;dim=5,\;it_{max}=100-500;\beta_{0}=1;\beta_{min}=0.2;\;\propto =0.5;\;\gamma =1$
FPA	$NP=20-40,\;dim=5,\;it_{max}=100-500;\;p=0.7$
NBA	$NP=20-40,\;dim=5,\;it_{max}=100-500;A$=0.5; r = 0.5; α = γ = 0.9; $f_{min}=0$; $f_{max}=1.5$
SCA	$NP=20-40,\;dim=5,\;it_{max}=100-500;$ a=[2−0]
SSA	$NP=20-40,\;dim=5,\;it_{max}=100-500;$ $c_{1}=\left[2-0\right]$
NMRA	$NP=20-40,\;dim=5,\;it_{max}=100-500;b_{p}=0.5$
NMRV 1.0	$NP=20-40,\;dim=5,\;it_{max}=100-500;b_{p}=0.5$
NMRV 2.0	$NP=20-40,\;dim=5,\;it_{max}=100-500;b_{p}=0.5$
NMRV 3.0	$NP=20-40,\;dim=5,\;it_{max}=100-500;b_{p}=0.5$

**Table 3 polymers-13-01702-t003:** Statistics of simulated algorithms with population size 40 over 30 independent runs for 500 iterations for Tensile Strength (*TS)* estimation.

Algorithm	Worst	Best	Average	Median	Std. Dev.
FFA	174.8324	174.9215	174.9123	174.9215	0.021354
FPA	174.9215	174.9215	174.9215	174.9215	1.20 × 10^−10^
NBA	174.9215	174.9215	174.9215	174.9215	1.17 × 10^−11^
SCA	174.61	174.9215	174.7647	174.757	0.143845
SSA	174.71	174.9215	174.897	174.9215	0.057414
NMRA	174.9215	174.9215	174.9215	174.9215	8.26 × 10^−14^
NMRV 1.0	174.9215	174.9215	174.9215	174.9215	8.67 × 10^−14^
NMRV 2.0	174.9215	174.9215	174.9215	174.9215	7.97 × 10^−14^
NMRV 3.0	174.9215	174.9215	174.9215	174.9215	8.24 × 10^−14^

**Table 4 polymers-13-01702-t004:** Statistics of simulated algorithms with population size 40 over 30 independent runs for 500 iterations for Flexural Strength (*FS)* estimation.

Algorithm	Worst	Best	Average	Median	Std. Dev.
FFA	162.6744	162.6744	162.6744	162.6744	5.78 × 10^−14^
FPA	162.6744	162.6744	162.6744	162.6744	5.78 × 10^−14^
NBA	162.1491	162.6744	162.6569	162.6744	0.095915
SCA	162.6744	162.6744	162.6744	162.6744	5.78 × 10^−14^
SSA	162.6744	162.6744	162.6744	162.6744	5.78 × 10^−14^
NMRA	162.6744	162.6744	162.6744	162.6744	5.78 × 10^−14^
NMRV 1.0	162.6744	162.6744	162.6744	162.6744	5.78 × 10^−14^
NMRV 2.0	162.6744	162.6744	162.6744	162.6744	5.78 × 10^−14^
NMRV 3.0	162.6744	162.6744	162.6744	162.6744	5.71 × 10^−14^

**Table 5 polymers-13-01702-t005:** Statistics of simulated algorithms with population size 40 over 30 independent runs for 500 iterations for Impact Strength (*IS)* estimation.

Algorithm	Worst	Best	Average	Median	Std. Dev.
FFA	1.605824	1.605824	1.605824	1.605824	2.26 × 10^−16^
FPA	1.605824	1.605824	1.605824	1.605824	2.26 × 10^−16^
NBA	1.605824	1.605824	1.605824	1.605824	2.26 × 10^−16^
SCA	1.605824	1.605824	1.605824	1.605824	2.26 × 10^−16^
SSA	1.605824	1.605824	1.605824	1.605824	2.26 × 10^−16^
NMRA	1.605824	1.605824	1.605824	1.605824	2.26 × 10^−16^
NMRV 1.0	1.605824	1.605824	1.605824	1.605824	2.26 × 10^−16^
NMRV 2.0	1.605824	1.605824	1.605824	1.605824	2.26 × 10^−16^
NMRV 3.0	1.605824	1.605824	1.605824	1.605824	2.26 × 10^−16^

**Table 6 polymers-13-01702-t006:** Statistics of simulated algorithms with population size 40 over 30 runs for 100 iterations for *TS* estimation.

Algorithm	Worst	Best	Average	Median	Std. Dev.
FFA	174.1897	174.9215	174.731	174.7742	0.194092
FPA	174.6641	174.9215	174.8728	174.9107	0.0739
NBA	174.6234	174.9215	174.8686	174.9215	0.112
SCA	174.4651	174.9213	174.6961	174.6229	0.140252
SSA	174.6239	174.9215	174.8157	174.877	0.117193
NMRA	174.9215	174.9215	174.9215	174.9215	4.31 × 10^−6^
NMRV 1.0	174.9215	174.9215	174.9215	174.9215	3.02 × 10^−7^
NMRV 2.0	174.9215	174.9215	174.9215	174.9215	5.48 × 10^−7^
NMRV 3.0	174.9215	174.9215	174.9215	174.9215	4.21 × 10^−8^

**Table 7 polymers-13-01702-t007:** Performance of simulated algorithms with population size 40 over 30 runs for 100 iterations for *FS* estimation.

Algorithm	Worst	Best	Average	Median	Std. Dev.
FFA	162.5976	162.6744	162.6555	162.6574	0.0187
FPA	162.615	162.6744	162.6697	162.6744	0.0152
NBA	162.1491	162.6744	162.6044	162.6744	0.181637
SCA	162.6744	162.6744	162.6744	162.6744	5.78 × 10^−7^
SSA	162.6744	162.6744	162.6744	162.6744	5.78 × 10^−7^
NMRA	162.6744	162.6744	162.6744	162.6744	6.65 × 10^−6^
NMRV 1.0	162.6744	162.6744	162.6744	162.6744	6.25 × 10^−7^
NMRV 2.0	162.6744	162.6744	162.6744	162.6744	2.25 × 10^−6^
NMRV 3.0	162.6744	162.6744	162.6744	162.6744	4.11 × 10^−7^

**Table 8 polymers-13-01702-t008:** Performance of simulated algorithms with population size 40 over 30 runs for 100 iterations for *IS* estimation.

Algorithm	Worst	Best	Average	Median	Std. Dev.
FFA	1.604239	1.605824	1.605749	1.605824	3.02 × 10^−4^
FPA	1.605824	1.605824	1.605824	1.605824	2.26 × 10^−10^
NBA	1.605824	1.605824	1.605824	1.605824	2.26 × 10^−10^
SCA	1.605824	1.605824	1.605824	1.605824	2.26 × 10^−10^
SSA	1.605824	1.605824	1.605824	1.605824	2.26 × 10^−10^
NMRA	1.605824	1.605824	1.605824	1.605824	2.52 × 10^−10^
NMRV 1.0	1.605824	1.605824	1.605824	1.605824	1.14 × 10^−10^
NMRV 2.0	1.605824	1.605824	1.605824	1.605824	9.56 × 10^−10^
NMRV 3.0	1.605824	1.605824	1.605824	1.605824	8.12 × 10^−11^

**Table 9 polymers-13-01702-t009:** Performance of simulated algorithms with population size 20 over 30 independent runs for 100 iterations for *TS* estimation.

Algorithm	Worst	Best	Average	Median	Std. Dev.
FFA	174.1762	174.9215	174.724	174.7416	0.189432
FPA	174.6634	174.9215	174.8716	174.9093	0.0727
NBA	174.6231	174.9209	174.8679	174.9128	0.159
SCA	174.4648	174.9213	174.6948	174.6218	0.150682
SSA	174.6236	174.9214	174.8149	174.8637	0.127843
NMRA	174.9214	174.9215	174.9215	174.9215	4.38 × 10^−5^
NMRV 1.0	174.9215	174.9215	174.9215	174.9215	3.13 × 10^−7^
NMRV 2.0	174.9215	174.9215	174.9215	174.9215	5.53 × 10^−7^
NMRV 3.0	174.9215	174.9215	174.9215	174.9215	4.52 × 10^−8^

**Table 10 polymers-13-01702-t010:** Performance of simulated algorithms with population size 20 over 30 independent runs for 100 iterations for *FS* estimation.

Algorithm	Worst	Best	Average	Median	Std. Dev.
FFA	162.218	162.6709	162.5367	162.5618	0.119
FPA	155.0595	162.6744	162.3814	162.3684	1.39
NBA	162.1491	162.6744	162.6219	162.6172	0.160298
SCA	162.1491	162.6744	162.6044	162.6059	0.182
SSA	162.6744	162.6744	162.6744	162.6744	5.78 × 10^−4^
NMRA	162.6744	162.6744	162.6744	162.6744	2.38 × 10^−5^
NMRV 1.0	162.6744	162.6744	162.6744	162.6744	1.02 × 10^−5^
NMRV 2.0	162.6743	162.6744	162.6743	162.6743	9.03 × 10^−5^
NMRV 3.0	162.6744	162.6744	162.6744	162.6744	1.72 × 10^−6^

**Table 11 polymers-13-01702-t011:** Performance of simulated algorithms with population size 20 over 30 independent runs for 100 iterations for *IS* estimation.

Algorithm	Worst	Best	Average	Median	Std. Dev.
FFA	1.602716	1.605824	1.605407	1.605812	7.42 × 10^−4^
FPA	1.599358	1.605824	1.605184	1.605806	1.74 × 10^−3^
NBA	1.605824	1.605824	1.605824	1.605824	2.26 × 10^−9^
SCA	1.605824	1.605824	1.605824	1.605824	2.26 × 10^−9^
SSA	1.605824	1.605824	1.605824	1.605824	2.26 × 10^−9^
NMRA	1.605823	1.605824	1.605824	1.605824	4.68 × 10^−8^
NMRV 1.0	1.605824	1.605824	1.605824	1.605824	2.22 × 10^−9^
NMRV 2.0	1.605824	1.605824	1.605824	1.605824	3.08 × 10^−9^
NMRV 3.0	1.605824	1.605824	1.605824	1.605824	1.71 × 10^−9^

**Table 12 polymers-13-01702-t012:** Process parameters selected for the fabrication of test parts.

Parameter	Details
Layer height	0.12 mm
Building orientation	0°
Raster angle	60°
Raster width	0.4064 mm
Air gap	0.008 mm
Infill pattern	Cubic
Infill density	50%
Printing speed	50 mm/s

## Data Availability

The data presented in this study are available on request from the corresponding author.

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
