# Peer review of "Optimization of FFF Process Parameters by Naked Mole-Rat Algorithms with Enhanced Exploration and Exploitation Capabilities"

_polymers, 2021, doi:10.3390/polym13111702_

Round 1

Reviewer 1 Report

The paper appears interesting but its form and presentation are poor. Especial the section of experimental validation.  Not many, but deep, amendments needed before it could be accepted. Please attend the following:

  1. The term FDM is not suggested, FFF is prefered.
  2. it takes to long to reach the main contribution in the abstract. a maximum of 2 lines are enough as introduction in the abstract. 
  3. Figure 1 is not needed. but if authors decided to keep it, fixed the following: arrows close to X- and Y-axis are crossing each other, avoid this. 
  4. Literature Review on how Naked Mole-Rats algorithms have been used in other mechanical engineering applications.
  5. variable "k" in equation 2 has not been defined.
  6. some variables appear in normal font type in the text, I believe these should be in itallics.
  7. parameter xshould be "building orientaiton"
  8. Why has the work been focused on strength and not stiffness? how can the algorithm be incorporated for stiffness.
  9. how are the strength results presented related to the bulck material properties?
  10. PArameters and bounds might be best to presented in a table.
  11. Page line 294, space between "least" and "10"
  12. add the units in the caption of the tables 2, 3, 4, 5, 6, 7, 8, 9 and 10. 
  13. There's no information provided the experimental part. what material was used? which machine? which are the manufacturing parameters used? Which are the machines used for mechanical testing? under which conditions? velocity of testing? how are the impact tests performed? 
  14. Diagrams of the samples, and the mechanical testing are neded to understand the experimental part. 
  15. what are the shape of the samples? mechanical properties of FDM-samples are highly dependent on the shape of the samples. 
  16. Plots should also include units.
  17. which was the infill pattern used to fill the samples? raster angle is not enough information. which was the infill density employed?

Author Response

19.05.2021

Dear Prof. (Dr.) Editor-in-chief,

Thank you for considering my manuscript titled Optimization of FFF Process Parameters by Naked Mole-Rat Algorithms with Enhanced Exploration and Exploitation Capabilities, for the publication in Polymers (MDPI). I am grateful to you and the reviewers for the valuable suggestions provided. I like to resubmit our manuscript by adding response to all your comments. Below please find the answers and actions taken to address these comments. All the suggestions are incorporated and highlighted with the yellow colour in manuscript. The locations of these changes have been mentioned, where possible, in the action points that respond to each reviewers’ comments. Here are the responses to the reviewer comments:

AUTHOR RESPONSE TO REVIEWER AND EDITOR COMMENTS

Manuscript ID: polymers-1226955

Paper title: Optimization of FFF Process Parameters by Naked Mole-Rat Algorithms with Enhanced Exploration and Exploitation Capabilities

The manuscript has been thoroughly modified and improved the quality of the content to meet the standards of the Journal. All the suggestions made by the learned referees are included in the revised manuscript. We are extremely thankful to the referees & editor(s) for their constructive comments and appreciation.

Response to Reviewer’s Comments

The authors are grateful to the reviewers for their suggestions that have all contributed to improving the manuscript.

The answers to reviewer’s comments are given in Tables below.

REVIEWER-1

The paper appears interesting but its form and presentation are poor. Especial the section of experimental validation.  Not many, but deep, amendments needed before it could be accepted. Please attend the following:

S.No.

Comments

Responses

1.       

The term FDM is not suggested, FFF is preferred.

As suggested the term FDM is replaced with FFF in whole manuscript

2.       

It takes to long to reach the main contribution in the abstract. a maximum of 2 lines are enough as introduction in the abstract. 

Some of the introductory sentences are removed

3.       

Figure 1 is not needed. but if authors decided to keep it, fixed the following: arrows close to X- and Y-axis are crossing each other, avoid this. 

The crossing of arrows in Figure 1 are removed.

4.       

Literature Review on how Naked Mole-Rats algorithms have been used in other mechanical engineering applications

Naked Mole-Rats algorithm (NMRA) is a relatively new optimization algorithm and has been successfully implemented for solving Antenna Design problems and wireless sensor networks localization problems. This is a first attempt to implement NMRA for solving manufacturing process optimization issues.

5.       

variable "k" in equation 2 has not been defined.

K represents the number of worker in sequence of worker’s group. The random solution ( th and th worker) chosen from the worker's group is  and . This is explained in Page 7 Line 211-212.

6.       

some variables appear in normal font type in the text, I believe these should be in itallics.

The font of variables has been changed to italics

7.       

parameter xshould be "building orientaiton"

Parameter x2 is renamed as suggested on Page 10 Line 273

8.       

Why has the work been focused on strength and not stiffness? how can the algorithm be incorporated for stiffness.

Stiffness determines the ability of part to restrict deformation. Literature survey indicates that, generally, the stiffness and tensile strength are dependent on same parameters in case of thermoplastics made by FFF. Hence, any one parameter can be selected as response. Moreover, stiffness can be calculated by Young’s Modulus E= Stress/Strain within elastic limits. The NMRA algorithm can be used for optimization of stiffness by considering the objective function of stiffness which can be calculated by aforementioned formula.

9.       

how are the strength results presented related to the bulk material properties?

The mechanical strength of FFF parts cannot be related to bulk material properties because due to layer-by-layer manufacturing, these properties significantly vary with building orientation, infill pattern, infill density, raster angle etc. Moreover, the anisotropic behaviour of ABS parts play a major role during strength testing. On the other hand, bulk properties of polymers do not depend upon these variables.

10.   

Parameters and bounds might be best to presented in a table.

As per the suggestions of Reviewer, the parameters and bounds are presented in Table 1.

11.   

Page line 294, space between "least" and "10"

The space is now introduced in least and 10. Page 11 Line 308

12.   

Add the units in the caption of the tables 2, 3, 4, 5, 6, 7, 8, 9 and 10. 

Sir, there is no need to add the units in the caption of the abovesaid tables, as these all are unitless quantities.

Worst, Best, Average, Median and Std. Dev. are the unitless quantities and are the output of the fitness values obtained using NMRA.

13.   

There's no information provided the experimental part. what material was used? which machine? which are the manufacturing parameters used? Which are the machines used for mechanical testing? under which conditions? velocity of testing? how are the impact tests performed? 

The discussion has been added. Materials used is ABS P400. FFF printer is Prusa I3. Manufacturing Parameters are displayed in Table 12. Universal testing Machine (UTM), supplied by Shanta Engineering Pvt. Ltd., Pune, India. Velocity of testing is 50 mm/min. UTM testing conditions are given in Page 18 Line 400-406. In present study, only tensile and flexural testing are performed for validation of NMRA

14.   

Diagrams of the samples, and the mechanical testing are needed to understand the experimental part

Sample drawings are given in Figure 8. The mechanical testing is shown in Figure 9c and 9d.

15.   

what are the shape of the samples? mechanical properties of FDM-samples are highly dependent on the shape of the samples. 

The samples are prepared as per ASTM standards. The shape and size of samples are shown in Figure 8. The internal structure of samples is cubic with 50% density.

16.   

Plots should also include units.

The plots are retrieved after execution of NMRA programs and hence, the output is unitless.

17.   

which was the infill pattern used to fill the samples? raster angle is not enough information. which was the infill density employed?

Infill pattern is “Cubic”, infill density 50% and raster angle 0.4064 mm. These details are also given in Table 12.

Reviewer 2 Report

The manuscript deals with the optimization of printing parameters in FDM. After a brief presentation of the state of the art (amendable to improve, in my opinion) the authors proposed new algorithms and discussed the obtained result in terms of the mechanical properties of the printed goods. The role of polymers in such work appears in the foreground (I do not know if it is positive for Polymers Journal). I have also some concerns about the decimal places used by the authors in the obtained data. My question is: are all of these decimal places justified? Which is the SD of each value? Considering you reported four to six decimal places, I have to wait for a very low SD. Furthermore, the English needs revision, I highlighted some suggestions, mistakes and typos to be corrected in the attached .pdf. I would suggest a minor revision before publication.

Author Response

19.05.2021

Dear Prof. (Dr.) Editor-in-chief,

Thank you for considering my manuscript titled Optimization of FFF Process Parameters by Naked Mole-Rat Algorithms with Enhanced Exploration and Exploitation Capabilities, for the publication in Polymers (MDPI). I am grateful to you and the reviewers for the valuable suggestions provided. I like to resubmit our manuscript by adding response to all your comments. Below please find the answers and actions taken to address these comments. All the suggestions are incorporated and highlighted with the yellow colour in manuscript. The locations of these changes have been mentioned, where possible, in the action points that respond to each reviewers’ comments. Here are the responses to the reviewer comments:

AUTHOR RESPONSE TO REVIEWER AND EDITOR COMMENTS

Manuscript ID: polymers-1226955

Paper title: Optimization of FFF Process Parameters by Naked Mole-Rat Algorithms with Enhanced Exploration and Exploitation Capabilities

The manuscript has been thoroughly modified and improved the quality of the content to meet the standards of the Journal. All the suggestions made by the learned referees are included in the revised manuscript. We are extremely thankful to the referees & editor(s) for their constructive comments and appreciation.

Response to Reviewer’s Comments

The authors are grateful to the reviewers for their suggestions that have all contributed to improving the manuscript.

The answers to reviewer’s comments are given in Tables below.

REVIEWER-2

Comments and Suggestions for Authors

The manuscript deals with the optimization of printing parameters in FDM. After a brief presentation of the state of the art (amendable to improve, in my opinion) the authors proposed new algorithms and discussed the obtained result in terms of the mechanical properties of the printed goods. The role of polymers in such work appears in the foreground (I do not know if it is positive for Polymers Journal). I have also some concerns about the decimal places used by the authors in the obtained data. My question is:

S.No.

Comments

Responses

Q1.

Are all of these decimal places justified? Which is the SD of each value? Considering you reported four to six decimal places, I have to wait for a very low SD.

Sir, In Tables 3-10, the statistics is the outcome of the NMRA algorithm to determine the fitness value of TS, FS and IS values. These table also show the Standard Dev. value of statistics. The Standard Dev. value is very low because all of these algorithms reach to the optimum value with very less margin. And results show that NMRA variants attain good results in comparison to competitive algorithms.

Q2

Furthermore, the English needs revision, I highlighted some suggestions, mistakes and typos to be corrected in the attached .pdf. I would suggest a minor revision before publication.

As per suggestions, all the mistakes and typos are now corrected.

The errors highlighted in PDF version are also checked and highlighted in yellow color.

As per suggestions received, the grammatical errors have now been removed and many sentences have been reframed so as to deliver clear interpretation in the revised manuscript. Additionally, the ‘pdf’ Premium Grammarly report of the final revised version of the article which apparently shows the Score attained of 99 out of 100 which furthermore clearly reveals the utmost quality of the English language and writing skills employed while preparing the revisions has also been enclosed as a supplementary file for your kind consideration.

Thanks, in anticipation

Regards,

Shubham Sharma

(Corresponding Author)

Round 2

Reviewer 1 Report

Thanks for attending the comments in the 1st revision.